# Brief report: Lymph node morphology in stage II colorectal cancer

Annabelle Greenwood[1]*, John Keating[2], Diane Kenwright[3], Ali Shekouh[2], Alex Dalzell[2], Elizabeth Dennett[1,2], Kirsty Danielson[1]

1 Department of Surgery and Anaesthesia, University of Otago, Wellington, New Zealand, 2 Department of General Surgery, Wellington Regional Hospital, Wellington, New Zealand, 3 Department of Pathology and Molecular Medicine, University of Otago, Wellington, New Zealand

* grean360@student.otago.ac.nz

## Abstract

### Background

Colorectal cancer is one of the leading causes of cancer-associated morbidity and mortality worldwide. The local anti-tumour immune response is particularly important for patients with stage II where the tumour-draining lymph nodes have not yet succumbed to tumour spread. The lymph nodes allow for the expansion and release of B cell compartments such as primary follicles and germinal centres. A variation in this anti-tumour immune response may influence the observed clinical heterogeneity in stage II patients.

### Aim

The aim of this study was to explore tumour-draining lymph node histomorphological changes and tumour pathological risk factors including the immunomodulatory microRNA-21 (miR-21) in a small cohort of stage II CRC.

### Methods

A total of 23 stage II colorectal cancer patients were included. Tumour and normal mucosa samples were analysed for miR-21 expression levels and B-cell compartments were quantified from Haematoxylin and Eosin slides of lymph nodes. These measures were compared to clinicopathological risk factors such as perforation, bowel obstruction, T4 stage and high-grade.

### Results

We observed greater Follicle density in patients with a lower tumour T stage and higher germinal centre density in patients with higher pre-operative carcinoembryonic antigen levels. Trends were also detected between tumours with deficiency in mismatch repair proteins, lymphatic invasion and both the density and size of B-cell compartments. Lastly, elevated tumour miR-21 was associated with decreased Follicle and germinal centre size.

**Data Availability Statement:** All relevant data are within the paper and its Supporting Information files.

**Funding:** This study was funded by the University of Otago, Wellington Dean's Research Grant (2017).

**Competing interests:** The authors have declared that no competing interests exist.

## Conclusion

Variation in B-cell compartments of tumour-draining lymph nodes is associated with clinico-pathological risk factors in stage II CRC patients.

## Introduction

In 2018 colorectal cancer (CRC) was the second leading cause of cancer-associated morbidity and third leading cause of mortality worldwide [1]. While clinical guidelines for treatment decisions are relatively straightforward for most patients, the decision to use adjuvant chemotherapy in patients with stage II disease is complicated [2]. Patients with Stage II CRC have a 5-year survival rate of 60–80%. However, additional survival benefit from adjuvant chemotherapy is only between 0–4%. High-risk clinicopathological features such as perforation, bowel obstruction, T4 stage and high-grade are associated with patients with 'higher-risk' stage II that may benefit from chemotherapy as patients with stage III do [2]. However, the current risk assessment has limited prognostic value where overall survival (OS) for 'low-risk' patients is still between 68–86% and for 'high-risk' patients between 57–76% [3, 4]. Further, this risk assessment lacks the ability to predict which subgroup will gain benefit from chemotherapy [5].

One possible reason for the observed clinical heterogeneity in this group is differences in the anti-tumour immune response. The tumour-draining lymph nodes (TDLNs) may be particularly relevant considering they serve either as effective barriers to tumour cell spread or as facilitators of dissemination of the primary tumour [6]. Uninvolved TDLNs function by allowing the expansion, differentiation and release of B and T cells by concentrating infiltrating tumour-associated antigens through their organised compartments including B cell follicles, germinal centres (GCs) and mantle zones [7]. Therefore, considering uninvolved TDLNs are the primary site for tumour antigen presentation and T cell/B cell activation [8], it is likely the type and extent of the TDLN-mediated immune response is important in tumour evolution in these patients. Despite this, histomorphological changes in TDLNs and their association with other clinicopathological risk factors have been poorly investigated in patients with stage II CRC.

Recently, there has been increasing interest in immunomodulatory microRNAs (miRNA) in tumour tissue. The expression of miRNA-21 (miR-21), which is arguably the most dysregulated miRNA in stage II tumours [9], functions to reduce the antigen-presenting capabilities of dendritic cells and suppresses anti-tumour T-cells in the tumour microenvironment (TME) [10–13]. Further, elevated miR-21 has been inversely associated with specific T cell populations within CRC [12] and has prognostic value in stage II [9, 14, 15]. Beyond the TME however, little is known about the distal effects of miR-21, even though tumourigenic T cell responses influence TDLN B-cell compartments [16] and miR-21 is known to travel within tumour-derived extracellular vesicles to more distal sites in the body [17, 18]. Taken together, the immunomodulatory effects of tumour markers such as miR-21 could be reflected in the histomorphology of TDLNs. Currently, little is known about the TDLN response with tumours that show high expression of miR-21.

The aim of this pilot study was to begin to characterise TDLN histomorphological features with tumour pathological risk factors including miR-21 in a small cohort of stage II CRC.

## Materials and methods

### Cohort

All participants in this study underwent surgery with curative intent at our institution between March 2017 and October 2018. Inclusion criteria were stage II adenocarcinomas of the colon or rectum where disease stage was defined according to the 8[th] Edition AJCC TNM staging system [19]. People with a history of malignancy, inflammatory bowel disease or those undergoing neo-adjuvant therapy were excluded prior to the analysis. A total of 23 patients were eligible for inclusion and all provided written informed consent. This study was conducted in accordance with the Declaration of Helsinki and ethical approval was obtained from the Health and Disabilities Ethics Committee, New Zealand (15/CEN/143; 18/CEN/138).

Demographic and clinicopathological data was extracted from hospital records and stored de-identified on a secured database (REDCap). The clinicopathological data included tumour T stage, cell type, grade, lymphovascular invasion, peri-neural invasion, multiple polyps, number of TDLN excised, pre-operative carcinoembryonic antigen (CEA) levels and deficiency in mismatch repair (dMMR) proteins. Patients with dMMR tumours had no positive immuno-histochemical staining of at least one of the following MMR proteins; MLH1, MSH2, MSH6 and PMS2. High/low CEA was determined by a 3.5 μg/L cut-off value (private communication). All of this data was obtained and reported according to standard practices in the hospital laboratory. All of the following assays were performed blinded to patient demographic and clinicopathological data.

### H&E staining lymph nodes

Formalin fixed paraffin embedded (FFPE) lymph blocks were obtained from the Department of Pathology following routine histopathological assessment. FFPE samples were then placed in pre-made tissue microarray (TMA) blocks (Unitma) as 3 mm diameter cores using a Manual Tissue Microarrayer (Quick-Ray[TM]). Representative areas from the donor blocks were chosen based on the density of tissue and embedded in TMA recipient blocks. Sections of 3 μm were cut in triplicate on a microtome (Sakura Tissue Tek) and transferred to adhesive-coated slides. Triplicate sections were air dried and incubated at 60 ˚C for 60 minutes. Sections were dewaxed with xylene, rehydrated with graded alcohol washes, and stained with Haematoxylin and eosin (H&E).

### Histomorphological analysis of lymph nodes

The following evaluation procedure was based on a previous methodology by Seidl et al. [7]. Morphometrical analysis was performed on 4 x (field of view) H&E TDLN images (S1 Fig). All TDLN compartments were annotated using ImageJ software [20]. Specifically, GCs and B-cell Follicles (characterised by an intense blue ring around GCs) were quantified (S1 Fig). Follicle and GC density were calculated as the average number of Follicles/GCs per lymph node (f/n and GC/n) and Follicle and GC size were calculated by averaging the circumference of the 3 largest Follicles/GCs in each lymph node. Lastly, the mantle zone was calculated as the average Follicle area minus the average GC area. Triplicate sections were analysed individually and averaged. Both total individual lymph node values and averages of all lymph nodes per patient are reported.

### Tumour miR-21 analysis

Tumour and adjacent normal mucosa specimens were resected as fresh tissue at the time of surgery. Surgical samples were available for 20 patients in the cohort. Samples were split in two

for storage in RNA*later* (Life Technologies) or formalin at 4 ˚C. Formalin fixed tissues were embedded in paraffin, sectioned and stained with H&E to ensure that the samples analysed were tumour. A single pathologist confirmed the H&E slides of abnormal tissue as tumour, validated the tumour histological type/grade and determined the tumour cell percentage (S2 and S3 Figs). The same pathologist also confirmed there were no tumour cells within normal tissue. One participant was excluded from the analysis where the abnormal sample could not be confirmed as malignant.

RNA was extracted from the tissue stored in RNA*later* using the miRNeasy kit (Qiagen) following the manufacturer's protocol. Total RNA quantity and quality was validated using the Nanodrop spectrophotometer and 10 ng of total RNA was used to synthesise cDNA using the TaqMan[TM] Advanced miRNA cDNA Synthesis Kit (Applied Biosystems). miRNA expression of hsa-miR-21-5p, hsa-miR-345-5p and hsa-miR-16-5p was examined by real-time PCR using the TaqMan[TM] Fast Advanced Master Mix, specific miRNA assays (Applied Biosystems) and the RotorGene 6000 detection system. The geometric mean of the two housekeeper genes (miR-345 and miR-16) were used to normalise miR-21 average $C_t$ values. The fold change for each sample was calculated using the $2^{-\Delta\Delta Ct}$ method [21] based on a comparison between matched tumour and normal mucosa. A fold change of $\geq 2$ was considered overexpression of miR-21.

## Statistical analysis

All statistical analysis was performed using GraphPad Prism software (GraphPad Prism 7.00 software, Inc.). Categorical variables differences were tested using Fisher's exact tests. Correlations between continuous variables were tested using Spearman Rank correlation. Normally distributed continuous variables were reported as mean ± standard deviation and differences were tested using unpaired t-tests. Non-normally distributed continuous variables were reported as median (interquartile range) and differences were tested using Mann-Whitney U-tests. A significance level of 5% was chosen.

## Results

### Cohort demographics and characteristics

A total of 23 participants were included in the analysis. Tumour samples were available for 20 out of 23 participants and TDLNs were available for 21 out of 23 participants. Of the entire cohort, the mean age was 65 years (range 34 to 80 years), and 15 participants were female (65%). Other cohort demographics and tumour characteristics are shown in Table 1. Participant demographic and tumour characteristics stratified by miR-21 and lymph node morphology status are available in S1–S3 Tables.

### Histomorphology of TDLNs

All TDLNs excised from surgical specimens in the 21 participants with stage II CRC were collected. TDLNs were excluded from the analysis if less than 50% of lymph node tissue was present inside the 3 mm cores. A total of 251 lymph node cores were examined for morphology patterns. For 13 participants at least 10 lymph nodes were examined while the remainder had between 3–9 lymph nodes examinable.

Histomorphological patterns were stratified by clinicopathological features (S2 and S3 Tables).

**T stage and Follicle/GC density.** Average Follicle density was significantly higher in patients with T3 tumours (14 ± 3 f/n) compared to patients with T4a/b tumours (11 ± 3 f/n,

**Table 1. Cohort demographics and tumour clinicopathological characteristics.**

| | | N (%) | Mean +/- SD (range) |
|---|---|---|---|
| **Gender** | Male | 8 (35) | |
| | Female | 15 (65) | |
| **Ethnicity** | European | 19 (83) | |
| | Māori | 2 (9) | |
| | Pacific Island | 1 (4) | |
| | Asian | 1 (4) | |
| **T stage** | T3 | 16 (69) | |
| | T4a | 5 (22) | |
| | T4b | 2 (9) | |
| **Histological type** | Adenocarcinoma | 19 (83) | |
| | Mucinous adenocarcinoma | 4 (17) | |
| **Grade (differentiation)** | Well | 15 (68) | |
| | Moderately | 1 (5) | |
| | Poorly | 6 (27) | |
| **CEA**[a] | <3.5 | 12 (52) | 1.84 ± 0.46 (1.1–2.4) |
| | ≥3.5 | 11 (48) | 11.54 ± 10.71 (4.1–32.4) |
| **Extramural vascular Invasion** | Yes | 1 (5) | |
| | No | 22 (95) | |
| **Peri-neural Invasion** | Yes | 1 (5) | |
| | No | 22 (95) | |
| **Lymphatic Invasion** | Yes | 5 (22) | |
| | No | 18 (78) | |
| **Multiple Polyps** | Yes | 10 (46) | |
| | No | 12 (54) | |
| **dMMR**[b] | No evidence | 13 (57) | |
| | Evidence | 10 (43) | |
| **TDLNs examined for cancer**[c] | ≥12 | 21 (93) | 25 (5–50) |
| | <12 | 2 (7) | |
| **TDLNs examinable for morphology patterns** | | | 11 ± 6 (3–25) |

[a]CEA level 3.5 μg/L cut-off (private communication).

[b]Deficiency in at least one of MSH2, MSH6, PMS2, MLH1.

[c]<12 resected lymph nodes is associated with a worse outcome [2].

$p = 0.03$; Fig 1A) and analysis by each individual lymph node reflected this association (14 (9–17) vs 10 (7–13) f/n, $p = 0.0002$; Fig 1B). Average GC density was also higher in patients with T3 tumours although this did not reach statistical significance (9 ± 3 vs 7 ± 4 GC/n, $p = 0.31$; Fig 1C). However, total individual lymph node GC density was significantly higher in T3 tumours (9 (5–14) vs 10 (7–13) GC/n, $p = 0.0005$; Fig 1D).

**dMMR and Follicle/GC density.** A statistical trend was detected between dMMR tumours and Follicle density, where patients with dMMR tumours had a higher average Follicle density ((14 ± 4 vs 12 ± 3 F/n, $p = 0.09$); Fig 1E). A strong association was also reflected in the individual lymph node analysis for Follicle density (14 (8–18) vs 11 (8–15) F/n, $p = 0.004$; Fig 1F). In patients with dMMR tumours there was also a trend towards a higher average GC density (10 ± 4 vs 8 ± 3, $p = 0.06$; Fig 1G) and higher total individual lymph node GC density (9 (5–14) vs 7 (4–11) GC/n, $p = 0.06$; Fig IH).

**CEA and Follicle/GC density.** Patients with high CEA levels (≥ 3.5 ng/ml) had non-significant higher average Follicle density compared to patients with low levels (12 ± 3 vs 14 ± 3

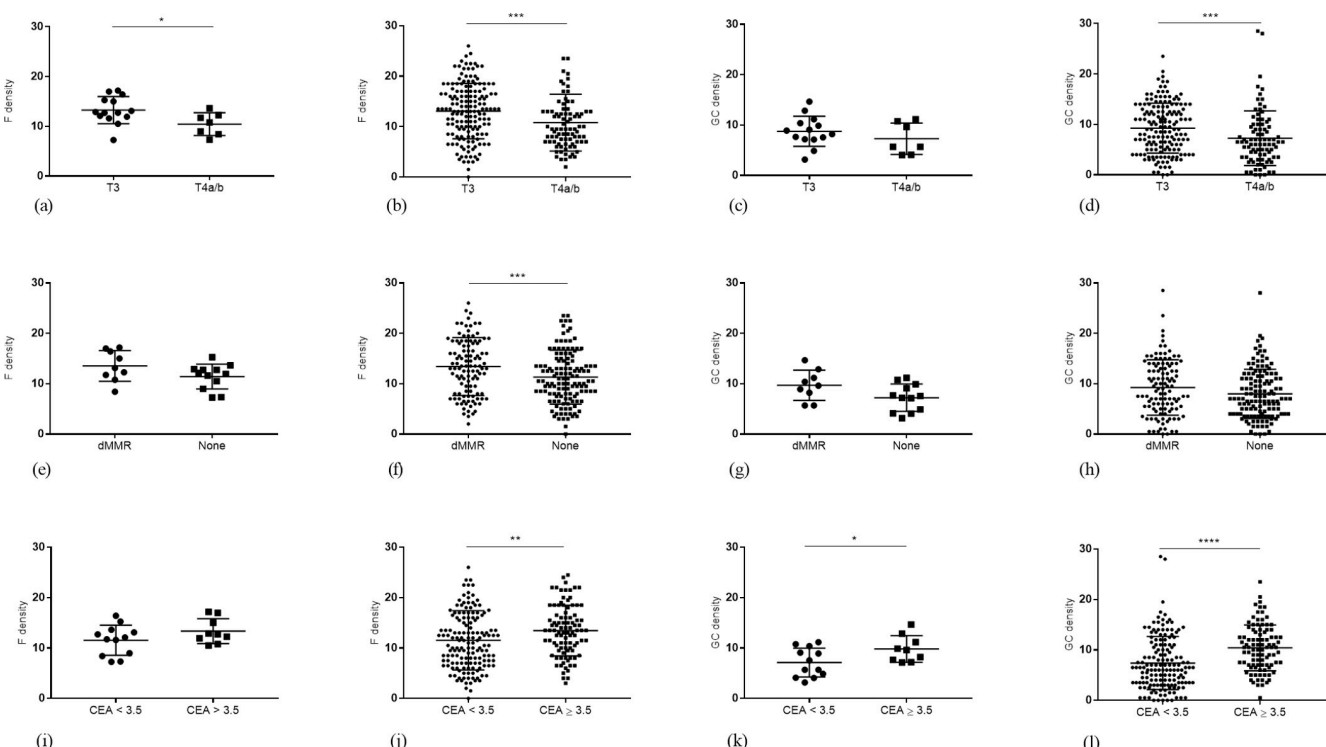

**Fig 1. TDLN histomorphology and clinicopathological features.** (A-D) B-cell Follicle and GC density by TDLN average and total individual TDLN in T3 versus T4a/b tumours. (E-H) B-cell Follicle and GC density by TDLN average and total individual TDLN in dMMR versus proficient MMR tumours. (I-L) B-cell Follicle and GC density by TDLN average and total individual TDLN in patients with serum CEA < 3.5 ng/ml versus serum CEA ≥ 3.5 ng/ml. f/ n = Follicles per node, GC/n = germinal centres per node. *p < 0.05, Student's t-test, Mann-Whitney U test.

F/n, $p$ = 0.15; Fig 1I). However, total individual lymph node Follicle density showed a strongly significant association by Follicle density (11 (7–6) vs 14 (10–17) F/n, $p$ = 0.003; Fig 1J). Average GC density reflected this association (8 ± 3 vs 10 ± 3 GC/n, $p$ = 0.04; Fig 1K) and total individual lymph node GC density was strongly significant (7 (4–11) vs 11 (7–14) GC/n, $p$ < 0.0001; Fig 1L).

Other associations detected between these TDLN morphology patterns and clinicopathological features included those between histological type, grade and lymphovascular/peri-neural invasion for Follicle/GC density or Follicle/GC size (S2 and S3 Tables).

### Tumour miR-21 and TDLN histomorphology

miR-21 expression in CRC tumour tissue and matched adjacent normal mucosa were compared in 19 out of 20 available samples. One sample was excluded from the analysis where abnormal tissue could not be confirmed as malignant. miR-21 tumour levels were 2-fold higher compared to normal mucosa (1.02 (0.89–1.1) vs 2.08 (1.1–2.8) $p$ = 0.0008; Fig 2A). In a comparison between tumour cell percentage and miR-21 tumour fold change, no correlation was detected ($r^2$ = 0.094, $p$ = 0.20; S2 Fig).

To understand whether miR-21 tumour levels were associated with histomorphological changes in TDLNs, high miR-21 expressing tumours were compared to low expressing tumours for TDLN histomorphological characteristics (S2 Table). A total of 17 out of 23 patients had both tumour miR-21 levels and TDLN morphology measures available for comparison. Patients with tumour miR-21 levels < 2 fold change had a significantly higher median

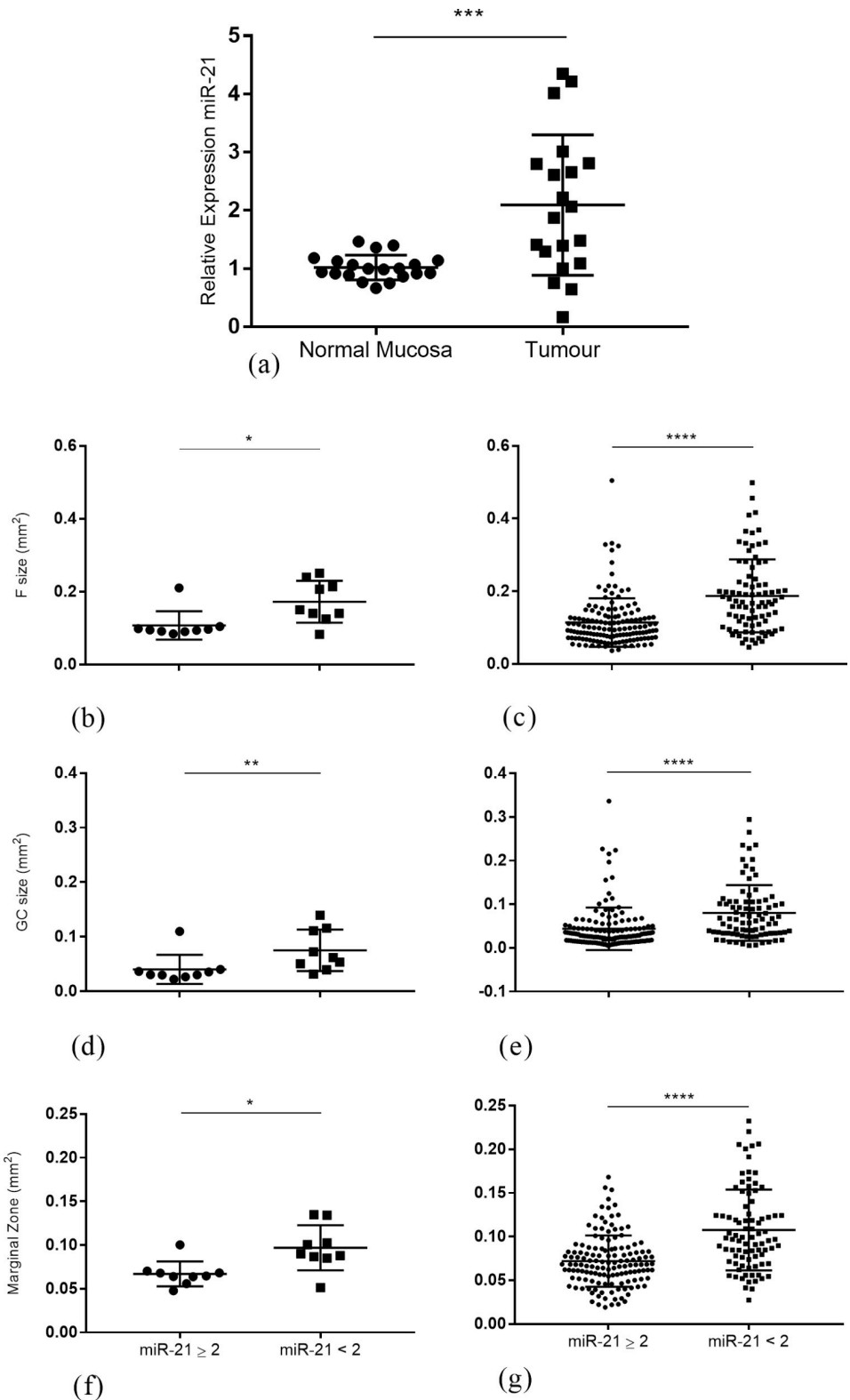

**Fig 2. TDLN histomorphology and miR-21 tumour levels.** (A) Relative expression of miR-21 in matched tumour and normal mucosa. (B, D, F) Median B-cell Follicle size, GC size and Marginal Zone in patients with tumour miR-21 ≥ 2 fold change compared to patients with tumour mi-21 < 2 fold change. (C, E, G) B-cell Follicle, GC size and Marginal Zone per lymph node in patients with tumour miR-21 ≥ 2 fold change compared to patients with tumour mi-21 < 2 fold change. ***p < 0.001, **p < 0.01, *p < 0.05, Student's t-test, Mann-Whitney U-test.

Follicle size (0.150 mm$^2$ (0.132–0.230) vs 0.094 mm$^2$ (0.091–0.102); $p$ = 0.03), GC size (0.062 mm$^2$ (0.045–0.110) vs 0.033 mm$^2$ (0.027–0.039); $p$ = 0.01) and Marginal zone (0.090 mm$^2$ (0.086–0.120) vs 0.064 mm$^2$ (0.058–0.070); $p$ = 0.01) compared to patients with miR-21 levels ≥ 2 fold change (Fig 2B, 2D and 2F). This was strongly reflected in the individual lymph node analyses for Follicle size (0.096 mm$^2$ (0.072–0.130) vs 0.170 mm$^2$ (0.110–0.230), $p < 0.0001$; Fig 2C), GC size (0.028 mm$^2$ (0.016–0.049) vs 0.066 mm$^2$ (0.034–0.106), $p < 0.0001$; Fig 2E) and Marginal Zone (0.068 mm$^2$ (0.052–0.086) vs 0.098 (0.073–0.128), $p < 0.0001$; Fig 2G).

## Discussion

This study is the first to observe associations between TDLN histomorphological features and tumour miR-21 levels in participants with stage II CRC. The key findings include: 1) Tumour T stage and serum CEA levels were associated with Follicle density and GC density, respectively: 2) Trends between dMMR tumours, lymphatic invasion and TDLN features were detected: 3) Elevated tumour miR-21 was associated with decreased Follicle, GC size and marginal zone.

Variation in the number and size of B cell primary Follicles and GCs in TDLNs could be reflective of a cancer-specific immune response and therefore provide prognostic information. These spatially organised compartments, along with T cell zones respond to the local tumour-associated immune signature within the draining lymph fluid [22]. Specifically, the immuno-modulation effects of B cell Follicles and GCs change the composition of B memory and plasma cell populations that home back to the site of the tumour [16]. While the effect of infiltrating immune cell populations is mostly focused on T cells, evidence is beginning to emerge for the role of infiltrating B cells [16]. Furthermore, with a clinically feasible approach in mind, B cell compartments are the most recognisable features that can be accurately quantified within TDLNs H&E slides; a process that could be automated in the future.

Our data suggest there is a potential link between TDLN histomorphological features and known tumour pathological risk factors. It is plausible that having a greater Follicle or GC density and size generates a stronger anti-tumour immune response. For example, dMMR tumours are known to be more immunogenic, characterised by an abundance of tumour infiltrating lymphocytes in response to MSI-induced frameshift peptides [23]. Our data indicate dMMR participants had a trend towards greater GC density, although this did not reach significance. Conversely, elevated tumour miR-21 is known to suppress the anti-tumour immune response in the TME and clinical studies suggest elevated miR-21 is associated with a poorer overall survival [9–15]. We show that elevated miR-21 is also associated with smaller average Follicle and GC size. While we have not shown any prognostic potential of the TDLNs, considering the various links to tumour pathological factors, this area is worth further investigation.

While no other groups have examined TDLN histomorphological features as biomarkers in CRC before, there have been insights into the TDLNs of patients with breast and oral cancer. Seidl et al. have reported that patients with stage I-III breast cancer, those with more aggressive and higher-risk breast cancer types had a significantly higher Follicle and GC density [7]. Conversely, Vered et al. found in a cohort of stage I-III oral cancer, a higher percentage of TDLN Follicles was associated with a significantly better prognosis [24]. The discrepancies in these studies and the overall lack of research in this area make it difficult to determine whether Follicle density actually reflects an anti-tumour response and how prognostic this is. Further, there is currently no consensus for quantifying TDLN histomorphological features and their prognostic value could be cancer type specific.

Limitations of the study include the lack of assessment of the prognostic ability of TDLNs in stage II CRC due to a small sample size and insufficient follow-up time. This area of

biomarker research is still well within the discovery phase and mechanistic studies are required to further elucidate the link between Follicle, GC density and an anti-tumour response as well as the role of tumour miR-21 in these processes. Validation of this work in a larger independent cohort is necessary to confirm the current findings. This may further inform the approach to quantifying TDLN histomorphological changes which is currently not standardised.

## Conclusions

In conclusion, we are the first to explore TDLNs histomorphological changes in patients with stage II CRC and their associations with tumour pathological factors. We suggest that the histomorphological variation could be reflective of cancer-specific host immune responses.

## Supporting information

**S1 Fig. Evaluation procedure for TDLN histomorphological analysis.** Evaluation is modified from a previously published method [7]. Left: All lymph nodes were annotated for B cell compartments (red and yellow dashed lines) using ImageJ. (I) Circular Germinal centre (GCs) within B-cell follicle, (II) Mantle zone. Right: Calculations for follicle and GC density, follicle and GC size, and mantle zone.
(TIF)

**S2 Fig. Correlation of miR-21 tumour fold change and tumour cells percentage.** $r^2 = 0.094$, $p = 0.20$, Pearson correlation.
(TIF)

**S3 Fig. Representative H & E tumour tissue slides.** Scale bar = 500 μM.
(TIF)

**S1 Table. Demographics, clinicopathological features and miR-21 expression.**
(DOCX)

**S2 Table. Demographics, clinicopathological features and TDLN histomorphology assessed by lymph node averages.**
(DOCX)

**S3 Table. Demographics, clinicopathological features and TDLN histomorphology assessed by individual lymph nodes.**
(DOCX)

## Author Contributions

**Conceptualization:** Annabelle Greenwood, Kirsty Danielson.

**Data curation:** Annabelle Greenwood, Diane Kenwright.

**Formal analysis:** Annabelle Greenwood, John Keating, Diane Kenwright, Ali Shekouh, Alex Dalzell, Elizabeth Dennett, Kirsty Danielson.

**Funding acquisition:** Kirsty Danielson.

**Supervision:** Elizabeth Dennett, Kirsty Danielson.

**Writing – original draft:** Annabelle Greenwood, Kirsty Danielson.

**Writing – review & editing:** Annabelle Greenwood, John Keating, Diane Kenwright, Ali Shekouh, Alex Dalzell, Elizabeth Dennett, Kirsty Danielson.

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
