## [Decision Letter · Decision Letter 0]

19 Nov 2020

PONE-D-20-25295

Brief Report: Lymph Node Morphology in Stage II Colorectal Cancer

PLOS ONE

Dear Dr. Greenwood,

Thank you for submitting your manuscript to PLOS ONE. After careful consideration, we feel that it has merit but does not fully meet PLOS ONE’s publication criteria as it currently stands. Therefore, we invite you to submit a revised version of the manuscript that addresses the points raised during the review process for additional data and analysis. 

We look forward to receiving your revised manuscript.

Kind regards,

Surinder K. Batra

Academic Editor

PLOS ONE

Journal Requirements:

Reviewers' comments:

Reviewer's Responses to Questions

**Comments to the Author**

1. Is the manuscript technically sound, and do the data support the conclusions?

Reviewer #1: Yes

Reviewer #2: Partly

2. Has the statistical analysis been performed appropriately and rigorously? 

Reviewer #1: Yes

Reviewer #2: No

3. Have the authors made all data underlying the findings in their manuscript fully available?

Reviewer #1: Yes

Reviewer #2: Yes

4. Is the manuscript presented in an intelligible fashion and written in standard English?

Reviewer #1: Yes

Reviewer #2: Yes

5. Review Comments to the Author

Reviewer #1: The Brief Report entitled “Lymph Node Morphology in Stage II Colorectal Cancer” by Greenwood et al. analyzed histomorphological changes and tumor pathological risk factors (e.g., the immunomodulatory microRNA-21 (miR-21)) in tumor-draining lymph nodes in a small cohort of stage II CRCs, with the overall goal of providing insight into the impact of anti-tumor immune response on the clinical course of stage II patients. Current risk assessment strategies for stage II CRC have limited prognostic value and lack the ability to predict which patients will benefit from chemotherapy. The study clearly shows that tumor T stage and serum CEA levels correlate with follicle density and germinal center density and that elevated levels of miR-21 are associated with decreased follicle and germinal center size and marginal zone, The study is well described and the data are clearly presented. The authors conclude that the number and size of B cell primary follicles and germinal centers may reflect cancer-specific host immune responses and may thus provide prognostic information. The data support these initial conclusions of the authors and encourage follow-up studies to assess the prognostic ability of tumor draining lymph node features in a larger cohort of stage II CRCs.

Reviewer #2: Greenwood et al provide brief report on variations observed in B-cell compartments of tumor-draining lymph nodes and their association with clinic-pathological risk factors in stage II CRC patient. Investigators observed correlative trends between tumors with deficiency in mismatch repair proteins, lymphatic invasion and both the density and size of B-cell compartments. Finally, elevated levels of miR-21 was found to be associated with decreased follicle and germinal center size. Though study finds association across various clinic-pathological features, however findings are just trend and no strong significant correlation were observed mainly due to limited number of cases. Further, no independent datasets is validated to confirm the findings. Mechanistic data showing an association of miR-21 with observed phenomenon is missing. It remains unclear how miR21 cause decreased follicle and germinal center size and what genes and pathways are dysregulated by it. With limited data and weak trend study findings will generate noise. Representative of histological findings need to be provided for each correlation observed across the study. It will be interesting to see how these correlation works in stage three and four cases. Further, considering correlation might not indicate causation, the study findings need to be reproduced in larger cohort of cases.

6. PLOS authors have the option to publish the peer review history of their article (what does this mean?). If published, this will include your full peer review and any attached files.

Reviewer #1: No

Reviewer #2: No

---

## [Author Response · Author response to Decision Letter 0]

19 Jan 2021

12th December 2020

Professor Surinda K. Batra

Academic Editor

PLOS ONE

Dear Professor Batra

Re: Submission of an original experimental study titled ‘Brief Report: Lymph Node Morphology in Stage II Colorectal Cancer’ (Manuscript ID PONE-D-20-25295). 

We thank the reviewers for their valuable and rigorous review and the journal for the opportunity to resubmit with the appropriate changes. We have responded to specific feedback below and hope that changes made to the manuscript will be sufficient for publication in PLOS ONE. 

Journal Requirements

We have adjusted our manuscript style, including file naming, to fit with PLOS ONE’s requirements. 

Reviewer 1

We would like to thank the reviewer for their positive review and response. 

Reviewer 2

1. ‘Though study finds association across various clinic-pathological features, however findings are just trend and no strong significant correlation were observed mainly due to limited number of cases.’

While we agree with the reviewer that our cohort size is small (n=23), this has consisted of the analysis of a total of 251 lymph nodes (range 3-25 per patient). To make this more clear, we have re-analysed the associations between lymph node morphology and clinicopathological features based on individual lymph nodes in addition to the mean values of a lymph node set per patient. Data presented in Figures 1 & 2 now shows both individual values and the mean value per patient. An additional supplementary table detailing results of all statistical analyses based on individual lymph nodes has also been included (Table S3) alongside the analyses based on the mean values of the lymph node sets per patient (Table S2). Information regarding these analyses have been added to the methods section (page 6, lines 130-131) and throughout the results section (pages 10-12). 

Statistical analysis of all data was performed using standard parametric (e.g. unpaired t-test) and non-parametric testing (e.g. Mann-Whitney U) as reported in the statistical analysis section (page 7). Multiple statistically significant correlations were found using p value of <0.05 for both the average lymph node values and individual lymph nodes. Regardless of this, we acknowledge that the small patient sample size is a limitation of this study, which we have ensured is stated on page 14 of the Discussion (lines 285-291): ‘Limitations of the study include the lack of assessment of the prognostic ability of TDLNs in stage II CRC due to a small sample size and insufficient follow-up time…. Validation of this work in a larger independent cohort is necessary to confirm the current findings.’ 

2. ‘Further, no independent datasets is validated to confirm the findings…. The study findings need to be reproduced in a large cohort of cases.’

We agree that the data requires further validation in an independent cohort; however, this is out of scope for this pilot study and we do not currently have an independent validation cohort available to us. This initial study was intended as an exploratory pilot to identify any associations and trends between lymph node histomorphological changes and tumour clinicopathological risk factors after observing large variances in these lymph node features in stage II patients. We are currently planning a larger scale prospective study based on these findings to validate our initial observations. This will allow us to limit the biases associated with retrospective studies and to collect additional data on the patient cohort studied, including outcomes such as time to recurrence and response to adjuvant treatments, which was not possible in this initial study. 

3. ‘Mechanistic data showing an association of miR-21 with observed phenomenon is missing. It remains unclear how miR21 cause decreased follicle and germinal center size and what genes and pathways are dysregulated by it.’

We agree with the reviewer that mechanistic studies investigating the association between tumour miR-21 and histomorphological changes to lymph nodes would be a valuable next step. We acknowledge that the data presented in this brief report is correlative only and we have shown no functional link between tumour miR-21 expression and lymph node histomorphology. This has been stated as a limitation in the Discussion on page 14 (lines 286-289): ‘This area of biomarker research is still well within the discovery phase and mechanistic studies are required to further elucidate the link between Follicle, GC density and anti-tumour response as well as the role of tumour miR-21 in these processes.’

We have instead speculated on the potential functionality of miR-21 in this setting based on the current literature which is discussed in the Introduction section (page 4, lines 67-75) and the Discussion section (page 13, lines 268-270). miR-21 represents one of the most intensively studied microRNA in CRC and is particularly well characterised as a suppressor of anti-tumour T cells in the CRC microenvironment. This knowledge was used to drive our aim of the study where we begin to characterise lymph node features with tumour miR-21 expression. While the observed association is interesting, an entire additional study would be needed to explore any mechanistic links between these associations. As this study was the first to report on histomorphological characteristics of tumour draining lymph nodes in CRC, we chose to focus on analysis of patient tissue samples and thus the study was not designed to carry out detailed mechanistic in vitro or in vivo work. This will instead be part of further follow up studies on this subject. 

4. ‘Representative of histological findings need to be provided for each correlation observed across the study.’

We apologise for the omission of these histological findings from the original manuscript. We have now included an additional supplementary figure (S3 fig.) of representative histological tissue samples. These samples were analysed by a single pathologist to confirm the histological cell type and percentage tumour cells. The normal mucosa samples were also confirmed to have no tumour cells present. The majority of histological findings including T stage, extramural vascular invasion, peri-neural invasion, lymphatic invasion and dMMR status were determined by pathologists in the hospital laboratory and reported according to standard protocols. These findings were extracted from medical records and we do not have access to the original representative images of all of these histological findings. 

5. ‘It will be interesting to see how these correlation works in stage three and four cases.’

We agree that it would be interesting to look at the tumour-draining lymph node features in stage III and IV patients. This may add some value to the findings of this study and we thank the reviewer for their insight. However, this study was exclusively a cohort of stage II patients with the ultimate goal of finding prognostic indicators in this patient group. We were particularly interested in stage II patients as there is no detectable tumour metastases to the lymph nodes at this stage. It is currently not possible to predict or detect pre-metastatic deposits in tumour-draining lymph nodes; however, changes to the histomorphology of the lymph nodes could be indicative of a pre-metastatic state or a local immune response to tumour-secreted factors. If this prognostic information was known this would benefit stage II patients as it could potentially change their treatment pathway. For stage III patients, metastatic tumour deposits are already detectable and additional prognostic information relating to lymph node morphology would be unlikely to change treatment pathways for stage III and IV patients. 

We would like to thank the editor and reviewers for their time and consideration on our revised manuscript. We hope that our responses to comments are acceptable and that our manuscript can now be considered of sufficient standard for publication in PLOS ONE. 

Sincerely,

Annabelle L. Greenwood

---

## [Decision Letter · Decision Letter 1]

15 Mar 2021

Brief Report: Lymph Node Morphology in Stage II Colorectal Cancer

PONE-D-20-25295R1

Dear Dr. Greenwood,

We’re pleased to inform you that your manuscript has been judged scientifically suitable for publication and will be formally accepted for publication once it meets all outstanding technical requirements.

Kind regards,

Surinder K. Batra

Academic Editor

PLOS ONE

Additional Editor Comments (optional):

Reviewers' comments:

Reviewer's Responses to Questions

**Comments to the Author**

1. If the authors have adequately addressed your comments raised in a previous round of review and you feel that this manuscript is now acceptable for publication, you may indicate that here to bypass the “Comments to the Author” section, enter your conflict of interest statement in the “Confidential to Editor” section, and submit your "Accept" recommendation.

Reviewer #1: All comments have been addressed

2. Is the manuscript technically sound, and do the data support the conclusions?

Reviewer #1: Yes

3. Has the statistical analysis been performed appropriately and rigorously? 

Reviewer #1: Yes

4. Have the authors made all data underlying the findings in their manuscript fully available?

Reviewer #1: Yes

5. Is the manuscript presented in an intelligible fashion and written in standard English?

Reviewer #1: Yes

6. Review Comments to the Author

Reviewer #1: (No Response)

7. PLOS authors have the option to publish the peer review history of their article (what does this mean?). If published, this will include your full peer review and any attached files.

Reviewer #1: No

---

## [Editor Report · Acceptance letter]

19 Mar 2021

PONE-D-20-25295R1 

Brief Report: Lymph Node Morphology in Stage II Colorectal Cancer 

Dear Dr. Greenwood:

I'm pleased to inform you that your manuscript has been deemed suitable for publication in PLOS ONE. Congratulations! Your manuscript is now with our production department. 

Kind regards, 

on behalf of

Prof. Surinder K. Batra 

Academic Editor

PLOS ONE